# Assessing the Cultural Ecosystem Services Value of Protected Areas Considering Stakeholders’ Preferences and Trade-Offs—Taking the Xin’an River Landscape Corridor Scenic Area as an Example

**DOI:** 10.3390/ijerph192113968

**Published:** 2022-10-27

**Authors:** Yue Su, Congmou Zhu, Lin Lin, Cheng Wang, Cai Jin, Jing Cao, Tan Li, Chong Su

**Affiliations:** 1College of Economics and Management, Anhui Agricultural University, Hefei 230036, China; 2Department of Land Resources Management, Zhejiang Gongshang University, Hangzhou 310018, China; 3College of Humanities and Foreign Languages, China Jiliang University, Hangzhou 310018, China; 4College of Environmental and Resource Sciences, Zhejiang University, Hangzhou 310058, China

**Keywords:** cultural ecosystem services, Q methodology, choice experiment method, preference, trade-off, sustainable management

## Abstract

Improving the accuracy of cultural ecosystem services (CESs) value assessment and paying more attention to the preferences and trade-offs of stakeholders in the administration of CESs are of vital importance for achieving resilient ecosystem management. Combining methodologies from sociology (Q method) and economics (choice experiment), an assessment framework of CESs is introduced to examine stakeholders’ preferences and willingness to pay to participate in CESs in protected areas so as to explore how the value of CESs in protected areas can be optimized. The results show that the selection of CESs by stakeholders reflects certain synergies and trade-offs. Visitors can be classified as preferring humanistic–natural recreation, aesthetic–sense of place, or environmental education according to the factor ranking of the Q method. Visitors have a higher willingness to pay for humanistic heritage and a lower willingness to pay for sense of place experience, which can be measured at $6.55 per visit and $0.96 per visit, respectively. This indicates that the local customs and characteristics should be further explored and promoted through traditional festival celebrations and farming activities in further development of protected areas, apart from protecting local cultural heritages such as Huizhou ancient villages and halls. Furthermore, it is also necessary to actively explore the synergistic development of CESs, promote social participation, raise stakeholders’ awareness of available services, manage visitors and stakeholders from a demand perspective, and promote the realization of the value of ecological products in protected areas.

## 1. Introduction

Nature reserves represent the last intact ecosystems worldwide and have a high potential to provide cultural and other ecosystem services [1,2]. The establishment of protected areas is an effective countermeasure to protect natural resources and deal with ecosystem degradation and biodiversity loss [3]. Protected areas not only provide material products for human survival, but also provide CES products to improve the quality of life and promote physical and mental health [4]. Billions of tourists visit the protected areas every year to appreciate the scenery, learn about natural and cultural heritage, and participate in local entertainment activities, socializing and learning, etc. Thus, protected areas provide a beneficial link between ecosystems and human well-being.

Linking changes in ecosystems to human well-being requires a trade-off approach that considers both tangible and intangible benefits [2]. In the past 20 years, numerous environmental protection plans, platforms, and classification frameworks have been implemented around the world to formulate scientifically effective conservation targets (such as the Millennium Ecosystem Assessment (MEA), Intergovernmental Platform on Biodiversity and Ecosystem Services (IPBES) [5], and Common International Classification of Ecosystem Services (CICES) [6]. The ultimate goal of these programs is to provide a framework for assessing the value of ecosystem services, which can put ecosystem service theory into practice and reduce value gaps caused by differences in assessment methods and political institutions. Emphasizing the connection between people and nature, the value assessment framework for ecosystem services provides an effective tool for protected ecosystem management and can be used to explore the win–win goals of environmental protection and social well-being [7]. As a framework for assessing the benefits that humans derive from nature and public services, ecosystem services have been continuously practiced and improved on a global scale.

MEA defines CESs as a source of spiritual enrichment, cognitive development, recreation, and aesthetic appreciation that people derive from ecosystems and as one of the key elements of nature–society interactions [3,8,9]. Thus, CESs can be seen as the outcome of trade-offs and synergies between different services and as a complex outcome of human–nature interactions in protected areas. This interaction means that CESs do not follow a simple linear one-way sequence from feature definition to services and the benefits people derive from them [10]. Analyzing the value of intangible CESs (e.g., nature experiences, ecotourism, environmental education, cultural heritage, sense of place, scientific research, etc.) compared to the value of more tangible and common ecosystem services (e.g., provision of timber, food, and other products) is a formidable challenge that requires the integration of social science and humanities perspectives [11,12]. In this context, many scholars have carried out numerous studies on CESs, which provide useful ideas and methods for the evaluation of CESs [10,13,14,15]. For example, some studies have quantified the spatial and temporal heterogeneity of CESs based on geographic information technology [16,17]; others have applied the revealed preference approach for evaluating the value of recreational services [18]. Other scholars believe that a trade-off is important for making more effective decisions among multiple CESs [3]. In addition, some studies from the consumer perspective show that the construction of infrastructure in protected areas is the key to economic use of CESs [19], and there may be potential opposing combinations of human preferences. The above research provides a good academic accumulation of evidence for the evaluation of CES, but the following aspects still need to be deepened. First, people’s perceptions and sociocultural differences affect their preferences for different CESs of protected areas based on attention restoration theory [9,20,21]. Additionally, the generation of economic profits in protected areas is affected by tourist consumption, thus, understanding the public perception and preference of CESs can help support the deployment and flexibility of management strategies [13,22,23]. Meanwhile, paying attention to the trade-offs and demands of stakeholders on CESs is also conducive for improving the accuracy of CES assessment and achieving resilient ecosystem management. Second, from the methodological perspective of cultural service evaluation, the value of cultural services involves the intersection of multiple disciplines, such as sociology, economics, management, and natural sciences [9]. However, most previous research uses a single-disciplinary approach or focuses on a subcategory of CES, with little consideration of the combination of multiple disciplines [24,25]. Therefore, it is necessary to integrate multidisciplinary qualitative and quantitative data into the CES analysis and decision-making process.

This study aims at contributing to the proposal of an analytical framework for evaluating CESs by combining methodologies from sociology (Q method) and economics (choice experiment), as well as exploring the preferences and trade-offs of stakeholders for different CESs in protected areas. In this regard, the specific goals of this study are to: (1) evaluate the stakeholder’s trade-offs for CESs and establish an evaluation framework for CESs, (2) quantify and calculate the nonmarket value of different CESs in the Xin’an River Landscape Corridor Scenic Area, and (3) discuss the policy implications for further management of protected areas. This study is conducive to revealing the trade-offs of CESs in protected areas and providing better protected area management strategies.

## 2. Study Area

The Xin’an River Landscape Gallery Scenic Spot is located in She County, Huangshan, in the southern part of Anhui Province (Figure 1). There are various types of topographies in the region, which is dominated by mountainous landforms. The climate is primarily subtropical monsoon, and, as such, Huangshan experiences four distinct seasons. The average annual temperature is 15 °C, and the forest coverage rate is 82.9%. The Xin’an River Landscape Gallery Scenic Spot is located within the world cultural and natural heritage site and global geopark, Huangshan Scenic Area, and is famous for the combination of unique natural scenery, Hui culture, and ancient villages and dwellings. Moreover, this scenic spot is located in the Xin’an River ecological compensation policy pilot site, which is the first horizontal ecological compensation site in China and which undertakes the important task of providing various services and products in the ecosystem of the protected area [26].

At present, the main way to visit the scenic spot is to visit the ancient villages along the route by boat, on the premise of not destroying the local ecological environment, carrying out a series of activities such as visiting ancient trees, touring Huizhou folk houses, learning ancient fishing methods, picking various loquats and citrus fruits, and tasting local specialties. The scenic spot also attracts tourists by carrying out various environmental protection activities, marathons, and folk culture festivals. However, most of the scenic spots are located in mountainous areas, with rugged terrain and inconvenient transportation. Most tourists travel independently by car. According to the ticket sales data of various tourism portals, the passenger flow at the scenic spot is approximately 20% of that of the larger Huangshan Scenic Area. Increasing the income of scenic spots to procure the true value of ecological products has become a problem faced by protected areas and scenic spot management departments.

## 3. Research Methods and Data Sources

### 3.1. Research Methods

The research uses the Q method from sociology to evaluate the stakeholder’s trade-off demands for CESs and to establish an evaluation framework for CESs. On this basis, the nonmarket value of CESs is calculated through the choice experiment method used in economics.

#### 3.1.1. Q Method and Design

The Q method is a nonmonetary preference-inducing method that provides a comprehensive assessment of prominent ecosystem services [27]. This method requires a small number of samples and mainly uses the Likert scale to quantify the attitudes of respondents. It provides insights into the range of opinions that exist about some topic within a sample population and how those opinions differ and converge. It can aid in analyzing the subjective behavior of actors accurately and is considered a solution to help resolve disputed policy issues [28]. Based on this, this paper uses the Q method to explore the trade-off demands of the stakeholders of nature reserves for the cultural service of the ecosystem.

The first step of the Q method is to collect Q statements to identify and define the broader CESs [29]. To ensure the comprehensiveness of the questionnaire and the soundness of the statement, in the early stage of the research, multiple dimensions of cultural services identified in the literature were used in the first draft. Then, various stakeholders were invited to make a Q statement, which could be in the form of self-expression or pictures. On this basis, 35 declarative sentences were selected as preliminary propositions for pre-investigation, and a total of 10 people, including Xin’anjiang scenic spot managers, tourism experts, local residents, and tourists, were invited to score and screen the propositions (assisted by pictures). The initial propositions were adjusted according to the pre-investigation, and some important propositions were added. The descriptions of some propositions were modified to make it easier for interviewees to understand and accept. According to the research needs of the Q method, the collected sentences were sorted and classified into five categories: aesthetics, sense of place, cultural heritage, natural heritage, and environmental education (Table 1). Finally, 30 statements were determined and randomly marked with serial numbers to form a Q sample. All statements and related photos were printed on 30 cards, and the cards were placed in random order (Figure 2).

The second step is to select respondents to identify the various perspectives of CES trade-offs and ensure stakeholder diversity [30]. In addition to the focus group discussions in the first step, participants were asked to forward the questionnaire to others who might disagree on the importance of CESs to prevent overly homogenous participants. It should be emphasized that the goal of this step is only to collect various opinions, including the importance of CESs, and the respondents selected after strategic sampling included guesthouse owners, local villagers, etc. Moreover, the conclusions drawn cannot be used as the basis for the Q classification, and follow-up steps are still needed.

The third step is the recruitment of participants for valuation exercises and a discussion of drivers of change. The implementation process was as follows: First, participants were asked to rank the ecosystem services identified in the first step. Figure 2 illustrates the structure of the ranking exercise, also known as Q-sorting. Participants were asked to choose the three most important types of CESs, followed by the next four, and so on until the entire Q-sorting was completed. The Likert scale was used to divide the attitudes of the respondents into seven levels (Figure 3). It is designed as a quasi-normal distribution ranging from −3 (strongly disagree) to 3 (strongly agree), where 0 means neutral or lacking salience either way. After a general understanding of each Q-statement, the participants put each card containing the statement on the Q-board in order from least agreement to most agreement. Thereafter, the number of cards at each rank was filled in the Q scale to complete the data collection.

Second, a brief follow-up interview was conducted with each participant to gather information about the ranking of CESs as well as drivers of change perceived to be influential to the participant’s ability to receive the “most important” CESs.

The last part is data processing and analysis [31]. Data analysis required factor analysis of all Q classes collected using PQ Method software 2.35, including factor rotation. This process allows for paring down the large number of unique popular perspectives into a limited number of typified viewpoints or archetypes. Each archetype is expressed with a factor array, which is a Q-sort defined by all those participants that load onto a particular factor. That is, each archetype is defined by those participants who hold similar viewpoints regarding the topic of interest. Then, the correlation matrix was subjected to factor analysis using PCA and rotated using Varimax to extract the eigenvalues and estimate the percentage of variance explained.

#### 3.1.2. Choice Experiment Method

The selection experiment method has been widely used in the fields of resource and environmental economics and ecological economics in recent years, and its theoretical basis is the new consumer theory and the random utility theory [14,32,33]. In view of the special attributes of CESs and the inability to describe their non-market value, this study uses the selection experiment method to evaluate the non-market value of the main CESs identified by the Q method.

The main consumers of CESs are tourists in protected areas, and their selection utility *U* is mainly composed of two parts: a definite term *V* and a random error term *ε*, which can be expressed as:*U_ij_* = *v_ij_* + *ε_ij_*(1)
where *U_ij_* represents the overall utility when tourist *i* chooses plan *j*, *V_ij_* represents the observable utility when respondent *i* chooses plan *j*, and *ε_ij_* represents the unobservable utility when respondent *i* choose plan *j*, that is, the random disturbance term.

Due to the existence of random disturbance terms, the behavior of tourists cannot be predicted perfectly. The choice probability is used to express that the respondent chooses the program *j* instead of the program *k* under all the alternative situations. The probability *P_ij_* at this time can be expressed as:(2)$$P_{ij}=P(U_{ij}>U_{ik})\;=P(V_{ij}+\varepsilon_{ij}>V_{ik}+\varepsilon_{ik}),\forall k\neq j$$
and the observable utility function *V_ij_* can be further simplified to a linear function. Its expression is:(3)$$V_{ij}=\mathrm{ACS}+\beta X\prime_{jn}\;$$
where *ASC* is the specific alternative constant term, which represents the benchmark utility of tourists when they do not choose the improvement plan. When the tourists choose the improvement plan, it is assigned a value of 0; when the tourists choose to maintain the status quo, it is assigned a value of 1, and when *ASC* is negative, it means that the tourists are more likely to choose the improvement plan. *X′_jn_* represents the *n*th attribute variable in the program *j*; *β* is the parameter matrix of the utility.

The traditional multinomial logit model has certain defects. The random disturbance terms *ε_i_* in this model obey the same distribution, and different *ε_i_* are independent of each other. The model relies on the assumption of the independence of irrelevant alternatives (IIA), which leads to certain biases in the research results. The model assumes that tourists’ individual preferences are homogeneous, but, in fact, there may be heterogeneity in tourists’ individual preferences. Based on this, this study adopts the random parameter logit model, which relaxes the constraints of the independent and identical distribution assumptions, considers that the coefficient of the explanatory variable is a random variable that obeys a certain distribution, and believes that tourists’ preferences are heterogeneous, which is more in line with reality. The formula for the probability *P_ij_* of tourist *i* choosing program *j* is as follows:(4)$$P_{ij}=\int \frac{\exp\left(V_{ij}\right)}{\sum_{k=1}^{j}\exp\left(V_{ik}\right)}f\left(\beta \right)d\beta$$

The parameters of each attribute are further estimated by the maximum likelihood estimation method, so that the marginal willingness to pay among tourists for each attribute of different CESs can be obtained. The formula can be expressed as:(5)$$\mathrm{WTP}=-\frac{\beta_{n}}{\beta_{p}}\;$$
where *β_n_* represents the estimated parameter of the *n*th attribute, and *β_p_* represents the estimated parameter of the tourist’s personal willingness to pay.

##### Attribute and Levels Design

The core of the choice experiment method design is to determine the attributes and set their different levels for participants to choose from for assessment. This paper refers to the relevant information from the Q method and combines the existing research to determine the following five cultural ecosystem service attributes:

Aesthetics mainly refers to the personal experience brought to tourists by the level of local infrastructure and environmental pollution [14]. The inappropriate development of protected areas for tourism may result in overcrowding, solid waste pollution, and noise pollution. Therefore, CESs in protected areas should pay attention to infrastructure and environmental governance. According by the previous research, three levels are set including deterioration, maintaining the status quo, and slight improvement.

The sense of place mainly refers to local customs and traditional culture, especially intangible cultural heritage including traditional crafts [10]. For example, since the study area has a long history of tea production, farming activities about picking tea can enable tourists to fully experience the heritage of tea culture in the Xin’an River Basin. Three levels are set including maintaining the status quo, slight improvement, and better improvement.

Cultural heritage mainly refers to the protection of local cultural relics and ancient dwellings. Known for its “ancient alleys, ancient walls, ancient bridges and ancient pagodas”, Shexian has the best-preserved ancient city of Huizhou and attracts many tourists every year [34]. In this regard, three levels are set including deterioration, maintaining the status quo, and slight improvement.

Natural heritage mainly refers to the rich natural landscapes and resources that provide excellent conditions for the development of local tourism [35]. As an important ecological reserve, the protection of the natural landscape in Xin’an River Basin is related not only to the sustainable development of local tourism but also to the ecological environment of the lower reaches of the Xin’an River. Based on this, the attributes of natural landscape protection are set to three levels: deterioration, maintenance of the status quo, and slight improvement.

The natural environment provides opportunities for education about pro-environmental values, attitudes, and behaviors, which can promote the formation of personal awareness of environmental responsibility, which leads to environmental behavior [36]. This paper sets three levels of environmental education, including maintaining the status quo, slightly improved, and better improvement.

Finally, this paper examines tourists’ willingness to pay for the above five types of attributes to measure the economic value and preference of CESs in protected areas. Referring to previous research, the respondents were asked what amount of public welfare fees they were willing to pay each time they participated under the assumption of receiving free tickets, and four levels were set, including $0, $7.61, $15.22, and $30.44 [37].

##### Orthogonal Design

The attributes and their state levels set in Table 2 are combined and 3 × 3 × 3 × 3 × 3 × 4 = 972 alternatives for attribute combinations are obtained. An orthogonal experiment is designed by using JMP software to reduce the difficulty of implementation [33]. Finally, 16 reasonable schemes were identified with a D-efficiency of 85.48%, indicating a good orthogonal effect. The status quo is used as the benchmark scheme, and together with 16 improvement schemes, 8 selection sets are finally obtained. Each choice set includes a benchmark scheme for maintaining the status quo and two improvement schemes. Table 3 shows a specific choice set example.

### 3.2. Data Sources

This article uses survey data from the Xin’an River Landscape Gallery Scenic Area acquired during 2019 and 2020. Due to the characteristics of the Q method, only a small sample (n = 30) is required for analysis. Therefore, a total of 35 questionnaires were distributed with the Q method, and 30 questionnaires were actually collected, with an effective rate of 85.7%. In the economic assessment of the nonmarket value of CESs, 800 questionnaires for the choice experiment were distributed, and 646 respondents completed the survey, representing an effective response rate of 80.75%.

## 4. Results

### 4.1. Q Method Results

#### 4.1.1. Participant Characteristics

This research conducted a Q study on tourists, experts in related fields, and scenic spot managers through interviews and preinvestigations. The 30 respondents to the survey came from different regions, including Huangshan, Hefei City, and Xuancheng City of Anhui Province, and Quzhou City and Yiwu City of Zhejiang Province.

#### 4.1.2. Factor Analysis and Factor Rotation Score

Following the factor rotation, the rotated factor loadings are calculated and flagged. Factor loadings are weights that are used to reflect the strength of the correlation between samples and factors. The ranking of factor loadings can be used to determine the classification of respondents and to test whether the factor loadings are meaningful. A Q-sort was deemed significantly loaded on a factor at *p* < 0.01 if its loading was greater than 2.581/√N = 0.471, where N = 30 is the number of statements or the Q-set, and 2.58 corresponds to the 99.5% threshold of a normal distribution.

Table 4 shows that a total of 25 respondents are associated with the first 3 factors, the ratio of which is 13:7:5, accounting for 83.3% of the total number. It can be considered that these three factors are representative. The five excluded responses may have been generated by uncertainty about sentence choice.

After factor analysis and factor rotation processing, this paper extracts three types of tourist types with significant differences in subjective attitudes. According to the factor scores, some sentences are selected for key analysis, and different factor types are named.

**Type 1**: Those who prefer humanities–natural recreation (Figure 4A). This type of visitor scores higher on propositions 19, 21, and 22. They prefer to get close to the natural environment and enjoy the beauty of nature, and they also like to visit the local traditional cultural heritage to feel the charm of nature through personal experience. They are more likely to have emotional resonance when they see the cultural–natural heritage of the scenic spot, while they do not expect to make much use of local environmental education and infrastructure.

**Type 2**: Those who prefer aesthetics–sense of place (Figure 4B). Tourists of this type may attach more importance to the construction of local infrastructure in scenic spots and give high scores to propositions 1 and 5. They believe that the development of tourist attractions should increase leisure, entertainment, and shopping places and hope to improve the quality of accommodations in tourist attractions. This type of tourist believes that the development of tourism must improve the level of local infrastructure.

**Type 3**: Those who prefer environmental education (Figure 4C). This type of tourist mainly hopes to receive some environmental education in the local area. Propositions 26 and 30 convey tourists’ yearning for environmental learning and education, hoping to understand the connection between local wild plants and natural scenery and emotions through learning. Such tourists have a strong interest in ecological trails and visiting Huizhou architecture, but they are not interested in statements such as environmental pollution in scenic spots. Based on the above analysis, this type of tourist is more inclined to understand CESs as a source of environmental education.

### 4.2. Choice Experiment Results

Through the sociological analysis of the Q method, we have basically understood the demands of stakeholders for CESs, but the value of CESs still needs to be measured in monetary terms by economic methods. This paper uses the choice experiment method to conduct field research in scenic spots and obtains 646 valid questionnaires.

#### 4.2.1. Sociodemographic Characteristics of the Respondents

The sociodemographic characteristics of the 646 respondents are shown in Table 5. The respondents had 56.7% aged between 25 and 45 years old and 3.3% aged above 65 years old; 52.8% females and 48.2% males; 13.3% at the postgraduate level, 50.0% with junior college or university degrees, and 23.3% with a high school education. Most respondents took a trip 1–2 times or 2–3 times a year, accounting for 46% and 30.4%, respectively. Their monthly income varied from less than $304.41 to more than $7610.35. The largest proportion stands at 43.3%, representing those with incomes between $761.04 and $1522.07, followed by 23.3% with incomes between $304.41 and $761.04. The characteristics of the survey samples basically conform to a normal distribution, indicating that the sample selection is basically effective.

#### 4.2.2. Random Parameter Logit Model

This paper uses Nlogit5.0 software to evaluate 464 selected experimental samples, for a total of 2784 (464 × 3 × 2) groups of observations. The random parameter logit model allows the coefficient of the observed variable to change randomly in the sample, so this paper chooses to build a random parameter logit model, and the model regression results are shown in Table 6. To prevent the problem of model convergence and facilitate the calculation of the implicit price of each attribute level, this paper takes the payment amount as a fixed parameter and uses the variable with a significant standard deviation coefficient in the regression variable as an alternative random parameter. Facility construction and farming culture experience are used as random parameters.

(1)The influence of ASC. According to the results, the coefficient of ASC is significantly negative at the 1% level, indicating that, compared with maintaining the status quo, tourists are more willing to improve the regional CESs and, thus, choose the improvement plan.(2)The influence of various CES variables. According to Table 6, all attribute coefficients are strongly significant at the confidence level of more than 5%, indicating that tourists have a strong willingness to improve support for cultural heritage, aesthetics, environmental education, natural heritage, and sense of place and can generate more attraction and bring better service experience. From the perspective of the standard deviation coefficient, infrastructure construction and agricultural cultural experience are both significant at the 5% level, further indicating the existence of heterogeneity in tourist preferences.

#### 4.2.3. Willingness to Pay

Table 7 shows that tourists have different levels of willingness to pay for each attribute level of recreational services. Tourists would pay the highest amount for the attribute of cultural heritage, which is $6.55/visit (Table 7), followed by natural heritage ($4.03/visit), and the payment amount for the sense of place is the lowest. This shows that when tourists participate in CESs, they prefer cultural heritage and natural heritage and are less interested in the experience of sense of place. The possible explanation is that compared with the choice of undeveloped sense of place experiential activities, tourists are more inclined to choose activities with relatively developed infrastructure for local natural and cultural heritage. Therefore, tourists’ willingness to pay for a sense of place is low. Overall, tourists’ willingness to pay for local cultural service products is approximately $16.86/visit, which is basically consistent with the existing research results.

## 5. Discussion

### 5.1. Stakeholders’ Preference and Trade-Off of Different CESs

Identifying the trade-off and preferences of stakeholders on CESs and improving the accuracy of CES value assessment is the basis of sustainable ecosystem management. By integrating methodologies from sociology (Q method) and economics (choice experiment), this paper conducts an assessment on the CESs of the protected area of the Xin’an River Landscape Gallery Scenic Spot.

The result of the Q method shows that the selection of CESs by stakeholders reflects certain synergies and trade-offs. Synergy is manifested in the synergy of human heritage and natural heritage, the synergy of aesthetics and sense of place, etc. The trade-off is in the choice of different forms of CESs. One possible reason for the “synergistic effect” of CESs is that different CESs exist in a state of co-supply, co-dependence, or social preference. This result reflects that the complexity of CESs is often nonlinear. The uncertain interactions and trade-offs between the many variables affecting CESs and their intra-service variables also illustrate the complexity of stakeholder interactions with protected areas [10]. The complexity here may be because the properties of ecosystems arise from dynamic or nonlinear interactions between components, leading to unpredictable patterns.

Tourists have the highest willingness to pay for cultural heritage when assessing the nonmarket value of CESs. This is because cultural heritage such as ancient villages and halls have profound historical value and provide an important chance for people to understand and experience the local Huizhou culture [25,38]. At the same time, visitors show the lowest willingness to pay for intangible CESs such as a sense of place. This indicates that the local customs and characteristics should be further explored, and more local cultural elements should be incorporated and promoted in local cuisine, festival celebration, and the experience of farming operations [39]. Furthermore, tourists’ willingness to pay for CESs is $16.86/visit, which is quite different from the existing local scenic spot ticket price ($22.53). Therefore, in the future management of protected CESs, it is necessary to improve the management framework of CESs and solve the problem of unbalanced supply and demand on the basis of meeting the basic needs of tourists. Further, the attributes of various products in CESs need to be defined, stakeholders’ awareness of CESs needs to be improved, and the realization of the value of ecological products needs to be fully implemented.

### 5.2. Policy Revelation for Management of Protected Area

Protected areas provide a wealth of CESs, making them highly regarded as one of the most dynamic socioecological systems on Earth [40]. Considering and accommodating multiple disparate stakeholder perspectives about what is important and what should receive scarce management and planning attention is paramount in the management of protected areas [32].

Our analysis shows that environmental education has not received much attention from tourists among CESs in protected areas. Therefore, focusing only on protecting the environment while neglecting other aspects of protected area management may be counterproductive. The ecosystem services provided by the natural and cultural heritage of protected areas can contribute to environmental protection by generating positive feedback on improved policy and management of protected areas. Furthermore, given the increasing alienation of humans from natural ecosystems and the resulting loss of emotional affinity for nature and associated declines in pro-environmental attitudes and behaviors [15], the promotion of various CESs in protected areas can help to break down dissatisfaction with nature and environmental protection.

For the Xin’an River Scenic Area and specific protected areas, it is important to understand the institutions and their respective scenic locations, focusing on the continuum between protecting biodiversity and meeting human needs as part of the natural experience. This requires a full understanding of the natural potential, needs, and capacity requirements of CESs in a particular scenic spot [41]. Protected area managers need to reposition opportunities for people to interact with nature, both by prioritizing the development of resources for these opportunities and by focusing on the stakeholder experience of visiting a particular protected area.

Based on this, several policy suggestions are proposed regarding protected area sustainability and management in the future. First, the model of protected area management should try to avoid “singularization” and “one size fits all”. For protected area management agencies and ecological scenic spots, it is necessary to understand the roles of agencies in balancing protected areas and the needs of tourists. In actual management, the trade-offs between multiple CESs should be considered, including the trade-offs between the tourist experience offered by CESs [42], revenue-generating potential, and the needs of environmental protection, each of which should be clarified. Second, opportunities and constraints for CESs within the scenic area need to be identified, thereby promoting synergies between environmental protection and the provisions of CESs. Managers need to incorporate complementary concepts of various types of CESs to achieve a combination of environmental protection and tourist support, with trade-offs across different protected area networks.

### 5.3. Strengths and Limitations

The ecosystem service’s research needs as much variety of methods as the system we want to analyze exists in complexity and values plurality [24,43]. This paper adopts a comprehensive analysis of methods from sociology and economics to clarify the values and perceptions of different stakeholders on CESs. Overall, the impact of qualitative and quantitative research benefits from the participation of expert and non-expert knowledge, which can complement and validate each other. In addition, mixed methods also make the evaluation more meaningful and make up for possible methodological deficiencies. However, our analysis still has limitations, since we did not take all the three value dimensions of ecosystem services (i.e., ecological, economic, and social) into consideration in ecosystem service assessments. With more data obtained in future, we will integrate natural science into the social sciences to contribute to the effort to develop ecosystem service assessments, increasing the potential that research will inform applied decision making in a more sustainable way.

## 6. Conclusions

This paper provides a framework for the assessment of CESs that integrates nonmonetary and monetary valuation methods, emphasizing the importance of the valuation of CESs as a tool for communication and engagement with local communities and stakeholders. The social assessment reveals the stakeholders’ preferences and trade-offs for CESs in the Xin’an River Landscape Corridor Scenic Area. Then, the economic assessment evaluates non-market values of CESs, which indicate that visitors have a higher willingness to pay for humanistic heritage ($6.55/visit) and natural heritage ($4.03/visit) than that of aesthetics ($3.56/visit), environmental education ($1.77/visit), and sense of a place ($0.96/visit). Tourists’ total willingness to pay for CESs is $16.86/visit, which is lower than the existing local scenic spot ticket price ($22.53). Our analysis suggests that the management of protected areas should consider and accommodate disparate stakeholder perspectives and handle the trade-offs between the tourist preferences, revenue-generating potential, and the needs of environmental protection. Sustainable management in nature reserves will be achieved if they actively explore the synergistic development of CESs, manage visitors and stakeholders from a demand perspective, and promote the realization of the value of ecological products in protected areas.

## Figures and Tables

**Figure 1 ijerph-19-13968-f001:**
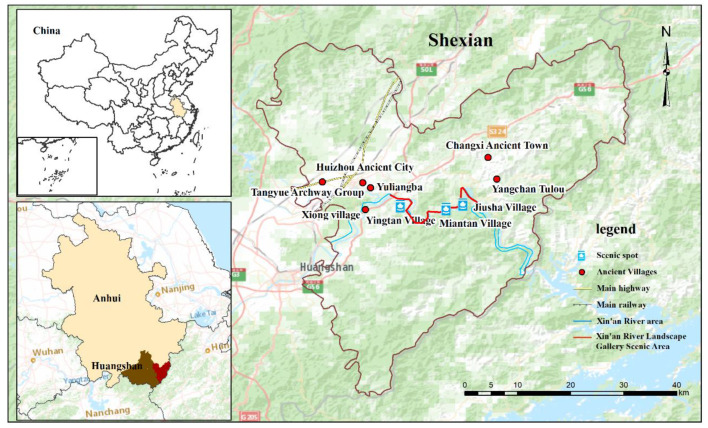
The location of the study area.

**Figure 2 ijerph-19-13968-f002:**
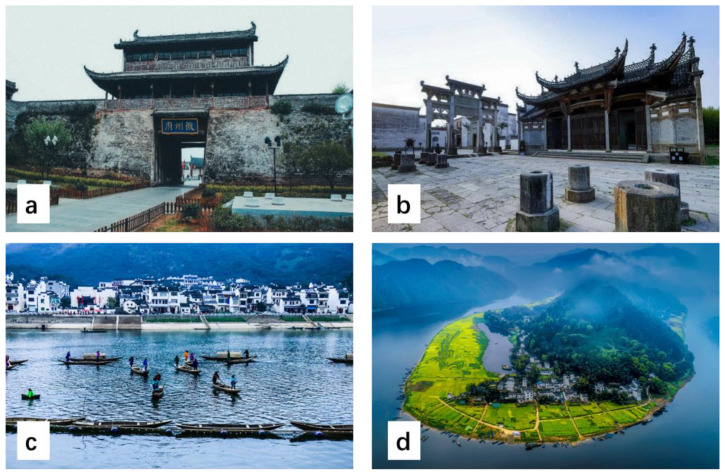
Some pictures used in the Q method. (**a**) Huizhou ancient city, (**b**) Huizhou ancestral hall, (**c**) traditional fishing culture, (**d**) natural scenery.

**Figure 3 ijerph-19-13968-f003:**
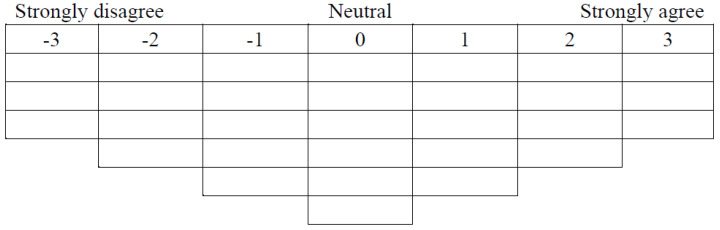
The structure distribution of the Q scale (N = 30).

**Figure 4 ijerph-19-13968-f004:**
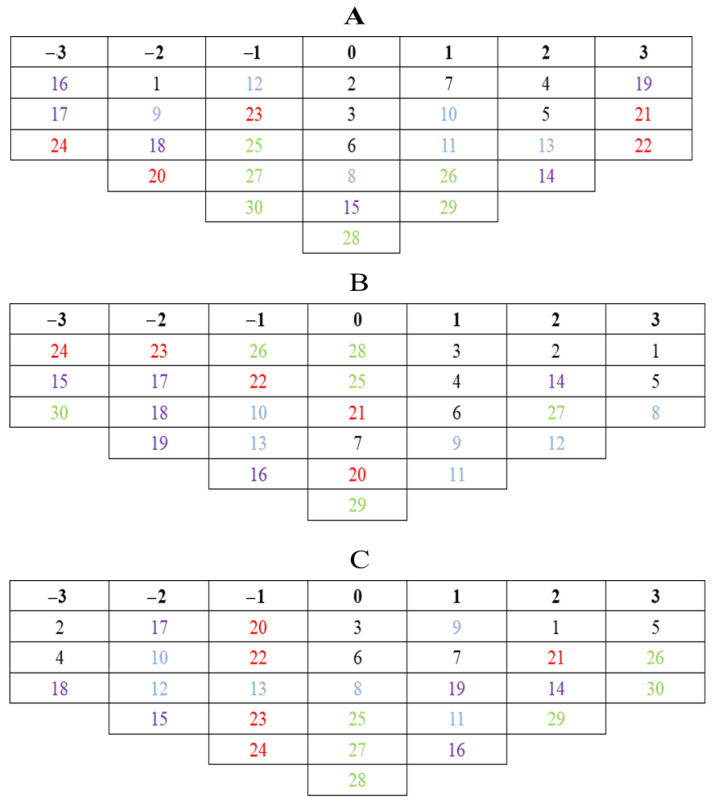
Types of tourists. And (**A**) represents preference for humanities-natural recreation; (**B**) represents preference for aesthetics-sense of place; (**C**) represents preference for environmental education. The color of black, blue, purple, red, and green represents aesthetics, sense of place, cultural heritage, natural heritage, and environmental education, respectively.

**Table 1 ijerph-19-13968-t001:** Q statements.

Category	Statement and Number	
Aesthetics: local infrastructure level	1	I hope accommodation in the scenic spot is good.
	2	I hope there is less solid waste in scenic spots.
	3	I hope the noise of the scenic spot is well controlled.
	4	I hope there is less water pollution in scenic spots.
	5	I hope there are more leisure and entertainment venues in scenic spots.
	6	I hope the scenic cruise is convenient.
	7	I hope the scenic road is accessible.
Sense of place: feel the local customs	8	Learn about traditional Huizhou culture (marriage, sacrifice, etc.).
	9	Learn about Hui merchants and tea appreciation.
	10	Taste local delicacies (stinky mandarin fish, hairy tofu, Huimo crisp sugar pastry, etc.).
	11	Learn about local history and traditions.
	12	Learn about local traditional handicrafts (Hui ink, She inkstone, bamboo carving, paper cutting, etc.).
	13	Learn about customs on farming and solar terms in Huizhou.
Cultural heritage: feel the local culture	14	Visit Huizhou architectural features (horse head wall).
	15	Visit Huizhou ancestral hall.
	16	Visit Huizhou ancient alley.
	17	Visit Huizhou ancient buildings.
	18	Visit Huizhou archway group.
	19	Visit a local ancient bridge.
Natural heritage: feel the natural scenery	20	Feel the traditional style of the local natural ecosystem.
	21	Go for recreational activities such as jogging on the ecological trail.
	22	Carry out a series of water-based activities such as cruises, etc.
	23	Go fishing.
	24	Go camping by the Sinan River.
Environmental education: carry out environmental knowledge learning	25	Learn about biodiversity.
	26	Learn about typical local wildlife.
	27	Learn about the main local plants.
	28	Participate in agricultural research activities.
	29	Learn green planting techniques.
	30	Learn about natural landscapes and emotional connections.

**Table 2 ijerph-19-13968-t002:** Cultural ecosystem service attributes and levels of the choice experiment.

Attributes	State Level	Attribute Hierarchy Explained
Cultural heritage	Deterioration	Local monuments are badly damaged and in disrepair.
	Status quo	There is some restoration of the damaged monument, but it is still in progress.
	Slightly improved	Ancient Huizhou architecture is repaired and protected, and ancient villages improved.
Aesthetics	Status quo	The local accommodation is average, and the transportation is not convenient.
	Slightly improved	There are chain hotels or B&Bs, and the transportation is relatively convenient.
	Better improvement	There are star hotels and special tourist buses, and the transportation is truly convenient.
Natural heritage	Deterioration	The original natural features are less preserved.
	Status quo	The original natural features are relatively preserved.
	Slightly improved	The original natural features are well preserved.
Environmental education	Status quo	There are no relevant environmental publicity and education efforts.
	Slightly improved	Some areas have environmental protection signs and environmental education.
	Better improvement	Most areas have environmental protection signs, environmental protection brochures, and environmental education.
Sense of place	Status quo	Tourists do not know there are local customs and experiential activities due to the lack of publicity.
	Slightly improved	There are some publicity signs, and tourists can sign up to experience custom activities such as farming and picking tea, etc.
	Better improvement	There are various forms of local custom activities, and tourists have a good sense of experience.
Willingness to pay	-	$0, $7.61, $15.22, $30.44 each year

**Table 3 ijerph-19-13968-t003:** Choice set example.

Attributes	Plan A	Plan B	Plan C
Cultural heritage	Status quo	Status quo	Status quo
Aesthetics		Status quo	Better Improvement
Environmental education		Status quo	Slightly improved
Natural heritage		Better Improvement	Status quo
Sense of place		Status quo	Status quo
Willingness to pay	$0	$7.61	$15.22
Your choice	-	-	-

**Table 4 ijerph-19-13968-t004:** Factor load table.

Sample	Factor Loading			Sample	Factor Loading		
	1	2	3		1	2	3
2	0.49 X	−0.18	−0.23	4	0.20	0.62 X	−0.09
5	0.88 X	−0.01	0.1	13	−0.07	−0.44 X	−0.10
6	0.71 X	−0.23	0.19	17	0.30	−0.66 X	−0.02
7	0.76 X	−0.30	0.11	18	0.31	0.55 X	−0.35
8	0.41 X	0.27	−0.10	25	0.30	0.63 X	0.26
9	0.37 X	−0.19	0.02	10	−0.21	0.04	0.60 X
11	0.80 X	0.16	0.16	16	0.18	0.04	0.58 X
14	0.63 X	0.14	−0.25	20	0.13	−0.13	−0.75 X
24	0.70 X	0.10	0.21	22	0.23	0.12	0.59 X
26	0.75 X	0.22	0.20	30	0.28	−0.16	0.74 X
27	0.79 X	0.19	0.25				
28	0.82 X	0.29	0.08				
29	0.45 X	0.20	0.23				
1	0.44	−0.49 X	0.02				
3	0.06	0.53 X	−0.01				

Note: The samples marked with X are the most representative in this category.

**Table 5 ijerph-19-13968-t005:** Descriptive statistics of the respondents.

Variable	Variable Definition	(% of Total Surveyed)	Mean	Standard Deviation
Gender	Male = 1 Female = 2	56.7	1.518	0.500
		43.3		
Age	25 years old and below = 1	16.7	2.139	0.819
	25–45 years old = 2	56.7		
	45–65 years old = 3	23.3		
	65 years old and above = 4	3.3		
Education level	Junior high = 1	13.3	1.909	0.919
	Senior high = 2	23.3		
	Undergraduate = 3	50.0		
	Postgraduate and above = 4	13.3		
Number of trips	0 times = 1	14.7	2.456	0.849
	1–2 times = 2	46.0		
	3–4 times = 3	30.4		
	4 times or more = 4	8.9		
Monthly income	$304.41 and below = 1	16.7	2.354	0.979
	$304.41–$761.04 = 2	23.3		
	$761.04–$1522.07 = 3	43.3		
	$1522.07–$7610.35 = 4	16.7		

**Table 6 ijerph-19-13968-t006:** Regression results of the random parameter logit model.

Variable	Coefficient	Standard Error
Random parameter		
Aesthetics	2.0132 ***	0.6674
Sense of place	0.5421 **	0.2366
Fixed parameters		
ASC	−2.3916 ***	0.6843
Cultural heritage	3.7076 ***	0.7529
Environmental education	1.00078 ***	0.3485
Natural heritage	2.2827 ***	0.6419
Payment amount	−0.08616 ***	0.0250
Standard deviation		
Aesthetics	0.7581 **	0.3715
Sense of place	1.8871 **	0.78018
AIC	1459.1	
Log likelihood	−715.5825	
Pseudo-R^2^	0.2255	

Note: ***, ** indicate that the estimated results are significant at the 1% and 5% levels.

**Table 7 ijerph-19-13968-t007:** Recreational value of each attribute.

Attributes	Willingness to Pay (Unit: $/Visit)	Sort
Cultural heritage	6.55	1
Aesthetics	3.56	3
Environmental education	1.77	4
Natural heritage	4.03	2
Sense of place	0.96	5
Total willingness to pay	16.86	-

## Data Availability

Not applicable.

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
