# Peer review of "Assessing the Cultural Ecosystem Services Value of Protected Areas Considering Stakeholders’ Preferences and Trade-Offs—Taking the Xin’an River Landscape Corridor Scenic Area as an Example"

_ijerph, 2022, doi:10.3390/ijerph192113968_

Round 1

Reviewer 1 Report

Dear Author,

Interesting paper that has a good potential. As is usually the case with reviews, I have some suggestions and comments for you authors. Please consider them and if you consider them important for this article (and your future articles) include.

Abstract must be improved. The abstract of the article must be written in correct English. The abstract is the most important part of the work as it is the part that will result in citing your article (not the article itself in most cases). Try to rewrite an abstract where Authors can refer to 4 important focused issues in approximately 200-250 words: (i) research motive, (ii) a void in the literature that makes this research important. (iii) test methodology; (iv) a summary of the findings. Overall, the abstract contained everything, but the English you use is difficult to understand, the sentences are too long.

Introduction can be improved. The introduction provides a good theoretical context for the study. A paragraph from line 77 contains stylistic mistakes- it is worth correcting.
It's also a good idea to add two paragraphs at the end of Introduction.
(1) First extensive paragraph in which you clearly outline the purpose of the research, problem (s), research procedure (schemat). It is worth considering the formulation of several hypotheses. This paragraph is a good place to write all of these points. This will make it easier for the reader to understand the entire context of the research.
(2) Last paragraph containing the structure of the article.

Paragraphs 2 and 3 fully describe the method and scope of the research.

Results. The results are well presented and described, both in the Q method results and in the survey results section. While the results of the Q method did not surprise me - the type of stakeholder of places like the Xianan River Landscape Corridor shares the group of those who appreciate humanities-natural rectration, or aesthetics-sense of place, or environmental eductation. Understandable. On the other hand, I was surprised by the regression results, especially in terms of willingness to pay. It would be worth somehow assessing the responses of the respondents - it is really difficult for people reading the results in Europe to assess the value of, for example, 43 yuan. You should consider this point, especially in relation to the citations of your article.

The article ends with a thoughtful discussion and recommendations.

Kind regards,

Author Response

Reviewer1

Dear Author,

Interesting paper that has a good potential. As is usually the case with reviews, I have some suggestions and comments for you authors. Please consider them and if you consider them important for this article (and your future articles) include.

Abstract must be improved. The abstract of the article must be written in correct English. The abstract is the most important part of the work as it is the part that will result in citing your article (not the article itself in most cases). Try to rewrite an abstract where Authors can refer to 4 important focused issues in approximately 200-250 words: (i) research motive, (ii) a void in the literature that makes this research important. (iii) test methodology; (iv) a summary of the findings. Overall, the abstract contained everything, but the English you use is difficult to understand, the sentences are too long.

Response: Thank you for the comment. We have revised the abstract of the paper according to your comment. Please see line 19-32.

Introduction can be improved. The introduction provides a good theoretical context for the study. A paragraph from line 77 contains stylistic mistakes- it is worth correcting.
It's also a good idea to add two paragraphs at the end of Introduction.
(1) First extensive paragraph in which you clearly outline the purpose of the research, problem (s), research procedure (schemat). It is worth considering the formulation of several hypotheses. This paragraph is a good place to write all of these points. This will make it easier for the reader to understand the entire context of the research.
(2) Last paragraph containing the structure of the article.

Response: Thank you for the comment. We have revised this part of the paper according to your comment. Please see line 63-95.

Paragraphs 2 and 3 fully describe the method and scope of the research.

Results. The results are well presented and described, both in the Q method results and in the survey results section. While the results of the Q method did not surprise me - the type of stakeholder of places like the Xianan River Landscape Corridor shares the group of those who appreciate humanities-natural rectration, or aesthetics-sense of place, or environmental eductation. Understandable. On the other hand, I was surprised by the regression results, especially in terms of willingness to pay. It would be worth somehow assessing the responses of the respondents - it is really difficult for people reading the results in Europe to assess the value of, for example, 43 yuan. You should consider this point, especially in relation to the citations of your article.

Response: Thanks for your thoughtful comments. We have changed the yuan to dollars to make it easier for worldwide readers to understand.

The article ends with a thoughtful discussion and recommendations.

Kind regards

Response: Thank you for your high recognition of this study. We are very appreciating your contribution for the quality improvement of our manuscript.

Reviewer 2 Report

This paper assessed the Cultural Ecosystem Services Value of Protected Areas Considering Stakeholders’ Preferences and Trade-Offs, which is an interesting work. However, it needs to be further improved in the following aspects.

(1) In the introduction part, the innovation of this paper need to be further clarified.

(2) There are many important ecosystem service areas in China. This paper takes Xin’an River Landscape Corridor Scenic Area as an Example, its representativeness and its reference value to the rest of the world need to be further explained. I would suggest summarize the research status more detail and specific the paper’s contribution in section 1.

(3) The perspective of this paper should be slightly changed. Its focus is too much centered on the case study, the broader perspective needs to be inquired in order to be interesting for the readers of this journal. Please collect and compare your case study with other similar cases- outside of China and possibly outside of Asia;

(4) In relation to the previous point, in the discussion of the results, it would be worthwhile to indicate the potential implications of the research in relation to other countries (referring to other local conditions).

(5) The conclusion needs to be refined. Conclusion should be the main finding of this paper.

(6) I think some figures about the questionnaires of Q-method could be added to increase the readability and make it well-presented.

(7)           Several minor errors need to be corrected:

Line 207: The “neutral” should be “Neutral”;

Line 208: The table 1 title should add a period;

Line 553-679: Some formatting issues about the references need to be revised.

Author Response

Reviewer 2

This paper assessed the Cultural Ecosystem Services Value of Protected Areas Considering Stakeholders’ Preferences and Trade-Offs, which is an interesting work. However, it needs to be further improved in the following aspects.

  • In the introduction part, the innovation of this paper need to be further clarified.

Response: Thank you for the suggestion. We agree with your point. We have revised this part of the paper according to your comment. Please see line 63-116.

  • There are many important ecosystem service areas in China. This paper takes Xin’an River Landscape Corridor Scenic Area as an Example, its representativeness and its reference value to the rest of the world need to be further explained. I would suggest summarize the research status more detail and specific the paper’s contribution in section 1.

Response: Thank you for the comment. First of all, we offer our sincere apologies that our explanation was not clear. We have revised this part of the paper according to your comment. Please see line 108-116.

(3) The perspective of this paper should be slightly changed. Its focus is too much centered on the case study, the broader perspective needs to be inquired in order to be interesting for the readers of this journal. Please collect and compare your case study with other similar cases- outside of China and possibly outside of Asia;

(4) In relation to the previous point, in the discussion of the results, it would be worthwhile to indicate the potential implications of the research in relation to other countries (referring to other local conditions).

Response: Thank you for the thoughtful suggestion. We have compared out study with other study in perspective of method and future implications in this paper. We have explained the potential implications of the research in section 5.2.

(5) The conclusion needs to be refined. Conclusion should be the main finding of this paper.

Response: Thank you for the comment. We have revised this part. Please see line 508-524.

(6) I think some figures about the questionnaires of Q-method could be added to increase the readability and make it well-presented.

Response: Thank you for the comment. We agree with your point. We have added Fig.2 in the manuscript (Please see line 179-181).

(7) Several minor errors need to be corrected:

Line 207: The “neutral” should be “Neutral”;

Line 208: The table 1 title should add a period;

Line 553-679: Some formatting issues about the references need to be revised.

Response: Thank you for the comment. We have revised these minor errors according to your comment.

Reviewer 3 Report

The manuscript is well written. Despite the coverage areas of non-valuation technique, this study has a broader scope to include another relevant issues of choice experiment. FGD driven attributes and their levels are always required under the choice experiment. This part is missing in this study. Additionally, the authors are failed to explain hypothetical bias and anchoring effect of the study. 

Author Response

Reviewer 3

The manuscript is well written. Despite the coverage areas of non-valuation technique, this study has a broader scope to include another relevant issues of choice experiment. FGD driven attributes and their levels are always required under the choice experiment. This part is missing in this study. Additionally, the authors are failed to explain hypothetical bias and anchoring effect of the study. 

Response: Thank you for the comment. We offer our sincere apologies that our description was not clear. When we conduct the questionnaires and review the data, we have confirmed the outliers and the respondents. In this investigation,the researcher confirmed the WTP value and record the reasons according to the large deviation of the value. Hence, we believe we control the bias and anchoring effect.